# Research on User Behavior Based on Higher-Order Dependency Network

**DOI:** 10.3390/e25081120

**Published:** 2023-07-26

**Authors:** Liwei Qian, Yajie Dou, Chang Gong, Xiangqian Xu, Yuejin Tan

**Affiliations:** College of Systems Engineering, National University of Defense Technology, Changsha 410073, China; qianliwei21@163.com (L.Q.); gcgc3737@163.com (C.G.); xuxiangqian18@163.com (X.X.); yjtan@nudt.edu.cn (Y.T.)

**Keywords:** higher-order dependency networks (HONs), behavior sequence analysis, random walk, vital node identification, community detection

## Abstract

In the era of the popularization of the Internet of Things (IOT), analyzing people’s daily life behavior through the data collected by devices is an important method to mine potential daily requirements. The network method is an important means to analyze the relationship between people’s daily behaviors, while the mainstream first-order network (FON) method ignores the high-order dependencies between daily behaviors. A higher-order dependency network (HON) can more accurately mine the requirements by considering higher-order dependencies. Firstly, our work adopts indoor daily behavior sequences obtained by video behavior detection, extracts higher-order dependency rules from behavior sequences, and rewires an HON. Secondly, an HON is used for the RandomWalk algorithm. On this basis, research on vital node identification and community detection is carried out. Finally, results on behavioral datasets show that, compared with FONs, HONs can significantly improve the accuracy of random walk, improve the identification of vital nodes, and we find that a node can belong to multiple communities. Our work improves the performance of user behavior analysis and thus benefits the mining of user requirements, which can be used to personalized recommendations and product improvements, and eventually achieve higher commercial profits.

## 1. Introduction

With the rapid popularization of the Internet of Things (IoT) and smart home, it has gradually become a mature way to collect user information through various sensors, cameras, and mobile phones to record users’ behavior data. User behavior analysis is of great significance to various business activities at present. According to the collected user behavior, the analysis results [1] can be applied to personalized recommendation services [2], product updates and improvements [3,4], and market demand research [5]. Accurate and reliable user behavior analysis results will be the key to achieve the above applications.

Typically, user behavior records are stored in the form of logs, which are processed to obtain time-stamped sequence data. Methods for analyzing sequence data include methods using machine learning [6,7,8] and methods using network analysis [9]. Machine learning methods mainly use models that support sequence data input, which can obtain higher accuracy but lack interpretability. Traditional network modeling and analysis methods are to build sequence data as an FON, and the edges of the network represent the relationship between nodes, which rely on the first-order Markov assumption, but the lack of higher-order dependency leads to lower accuracy. For user behavior analysis, it is necessary to have both high accuracy and interpretability to ensure the usability of the analysis results.

An HON is better able to mine historical information, and higher-order nodes are also able to show the sources of historical information [10]. An HON has higher accuracy improvement on sequence data with richer historical information and strong dependency [11], such as citation network data [12,13], trajectory data [14], etc. And there are also quite a few dependencies between user behaviors. Let us take an example; the case is shown in Figure 1.

In this case, an FON is constructed based on two kinds of sequences. The probabilities of transferring from *grasping onto a doorknob* node to *closing a door* and *opening a door* are 0.4 and 0.6, respectively. But, according to common sense, if you *open a door* before *grasping onto a doorknob*, you will not be able to *open a door* again, which means the action before *grasping onto a doorknob* will affect the transition probability distribution of subsequent actions. When building an HON, the *grasping onto a doorknob* node is split into two higher-order nodes, *grasping onto a doorknob|opening a door* and *grasping onto a doorknob|closing a door*. At this time, the network is split into two disconnected components. At this time, if *opening a door* before *grasping onto a doorknob*, the node status at the moment is higher-order node *grasping onto a doorknob|opening a door*, then the probability of *closing a door* in the next step is 1, and the probability of *opening a door* is 0, which is realistic. In HONs, due to the consideration of higher-order dependencies, some edges (the gray links in the case diagram) that exist in FONs will be removed. By removing these inaccurate edges, the accuracy of the random walk will be improved. Other random-walk-based algorithms will also benefit from it.

Therefore, considering the dependencies between user behaviors, in order to improve the accuracy of user behavior analysis and ensure interpretability, this paper will use an HON to model user behavior sequence data, and carry out random walk prediction, vital node identification, and community detection on the HON. The main contributions of this paper are as follows:We take the dataset Charades in the field of video behavior detection and the behavior label of each video in Charades [15] and CharadesEgo [16] as the dataset, and obtain the behavior sequence data through preprocessing data.We use the BuildHON+ algorithm to extract the higher-order dependency rules and build the HON of user behaviors.On this basis, we use RandomWalk, PageRank, and MapEquation algorithms to analyze the HON of user behavior, and compare the results in the FON, which shows the advantages of the HON over the FON.

The rest of the paper is organized as follows: Section 2 presents recent research related to behavior analysis and HONs. Section 3 introduces the first-order and higher-order Markov process. Section 4 introduces the HON and its construction methods. Section 5 conducts experiments for applying the HON to behavior data. Section 6 summarizes our work and discusses future directions. The structure of the paper is shown in Figure 2.

## 2. Related Works

Behavior analysis/valuation is a concept derived from psychology and medicine, which is used to predict and treat diseases [17]. Behavior analysis includes many applications and methods, some of which are carried out from the medical view [18], and some of which use data mining methods, including wavelet neural network [19] and recurrent neural network [20]. On the basis of various methods, behavior analysis is also widely used in other fields, such as abnormal behavior detection in Internet DNS servers [21], artificial intelligence training of automatic driving [22], and anomaly detection of user behavior [23]. Behavior analysis in the medical field is a process of causal inference, historical experience, and data analysis to detect potential diseases. This model is also applicable to requirement mining based on behavior analysis. Requirement mining based on behavior analysis is applied to the prediction of shared vehicle demand [24], intelligent prediction of medical demand [25], etc.

By reconstructing behavior sequence data into network, network science methods can be introduced into the study of behavior sequence data mining. Currently, most of the complex network modeling and reconstruction methods are based on traditional FONs [10,13]; however, in real life, higher-order interactions and higher-order dependencies are ubiquitous [26]. Collaboration among more than two authors of a paper [27], online social networks with group relationships among more than two people [28], etc., are all examples of higher-order interactions, which can generally be modeled as hypergraph or simplicial complex [29,30,31,32,33]. Networks that take these higher-order relationships into account are generally referred to as higher-order networks.

There are many kinds of high-order relationships between network nodes, and dependency relationship is one of them. Hypergraphs and simplicial complexes in common high-order networks are not suitable for describing dependencies. In order to describe higher-order dependencies on the network, the concept of a higher-order dependency network (HON) with memory nodes arises. The dynamics on a traditional FON are implicitly based on the first-order Markov assumption that the choice of the next target node depends only on the current node, but, in fact, all nodes before the current node may have an impact on the next transition [34].

Previously, HONs have been applied to data mining of sequence data. Rosvall et al. [34] improved the accuracy of algorithms such as PageRank and MapEquation by considering fixed second-order dependency, thus enabling the accuracy of community detection, ranking, and dynamics to be improved, and this effect was validated on several datasets. Xu et al. [10] released the limitation of fixed second-order dependency and extended the dependency to propose BuildHON to build a variable-order network, which was used in mining global shipping data to verify the advantages of the algorithm in terms of accuracy, scalability, and compatibility. Saebi et al. [14] improved BuildHON in terms of computational efficiency and proposed BuildHON+, which explored the application to anomaly detection on taxi trajectory data and applied it to HON embedding in subsequent work [35]. In addition, there are other algorithms for building HONs, such as GrowHON [36], ReHON [12], and SF-HON [37].

Existing behavior analysis methods are mainly based on FONs and machine learning methods which are hard to interpret, and no research has yet introduced higher-order Markov processes into behavior analysis. Our work considers this for the first time and uses an HON on user behavior sequence data, thus taking into account the dependencies between user behaviors.

## 3. Markov Process Considering Order

The first-order Markov assumption only considers the current state, but as the order increases, the historical state is gradually taken into consideration. The order of the HON is essentially the same as the order of the higher-order Markov process. The HON integrates the historical information in the higher-order Markov process into the higher-order nodes. Therefore, the construction of high-order dependency networks is based on the theory of high-order Markov processes. This section describes the Markov process considering order, mainly to provide the evolution from the first-order to the higher-order Markov process. In addition, the calculation of the state transition probability of the high-order Markov process is also the basis for the construction of the link weights between nodes in the HON. Therefore, it is necessary to describe the high-order Markov process before constructing the HON.

### 3.1. First-Order Markov Process

In a complex system of stochastic processes, there exists a sequence of states $\left\{i_{t},t=1,2,\cdots ,n\right\}$. $\left\{X_{t},t=1,2,\cdots ,n\right\}$ is a sequence of random variables that varies with time *t*. Each specific variable corresponds to a state in the system. With the flow of time, there is a transition between the states of the system, which can be described by the sequence of states $i_{0},i_{1},\cdots ,i_{n}$, and the corresponding sequence of variables is $X_{0},X_{1},\cdots ,X_{n}$. Typically, the probability of reaching state *i* at moment t+1 depends on all the previous states passed through, that is, the entire path history. It can be expressed as
(1)$$P(X_{t+1}=i_{t+1}|X_{t}=i_{t},X_{t-1}=i_{t-1},\cdots ,X_{1}=i_{1}).$$

However, with the advancement of the state transition process, the influence of the historical state farther from the current state on the prediction of the state transition probability at the next moment is significantly reduced, and retaining too much historical information will also cause a large amount of data redundancy and computing power waste. In many cases, the probability of transitioning to the next state depends only on the current state.
(2)$$\begin{array}{c} P(X_{t+1}=i_{t+1}|X_{t}=i_{t},X_{t-1}=i_{t-1},\cdots ,X_{1}=i_{1})\\ =P(X_{t+1}=i_{t+1}|X_{t}=i_{t}). \end{array}$$

A process that satisfies the above conditions is called a Markov process. Strictly speaking, it is the **first-order Markov process (M1)** [34]. The above probabilities are called transition probabilities and reflect only one-step transitions. When the number of transition steps is *k*, the k-step transition probability is defined as
(3)$$p\left(i_{t}\to i_{t+k}\right)=P\left(X_{t+k}=i_{t+k}|X_{t}=i_{t}\right).$$

In the **M1**, a series of state nodes and edges form a network and the weight of edges represent the transition probability distribution between state nodes. Therefore, the transition probability from state *i* to state *j* is
(4)$$p\left(i\to j\right)=\frac{W(i\to j)}{\sum_{k}W(i\to k)}.$$

We use W(i→j) to represent the weight of the edge from state *i* to state *j*. Based on the transition probability, and the probability at the previous state *i*, the probability of being in state *j* at the moment t+1 is
(5)$$P\left(j;t+1\right)=\sum_{i}P(i;t)p(i\to j).$$

### 3.2. Higher-Order Markov Process

The M1 supports most research on complex networks, but there are still many defects. The biggest problem is the assumption that the next state depends only on the current state. This assumption reduces the complexity of the algorithm and the redundancy of the data well, but it is not applicable to all situations. In our work, the M1 loses a considerable part of historical information and cannot capture the dependencies. Therefore, more historical information and path dependency need to be added, which is the higher-order Markov process. First, we start with the **second-order Markov process (M2)** [34].

In an **M2**, the next state depends not only on the current state, but also on the previous state.
(6)$$\begin{array}{c} P(X_{t+1}=i_{t+1}|X_{t}=i_{t},X_{t-1}=i_{t-1},\cdots ,X_{1}=i_{1})\\ =P(X_{t+1}=i_{t+1}|X_{t}=i_{t},X_{t-1}=i_{t-1}). \end{array}$$

Therefore, it is not the current node that decides the probability of the next state, but the path formed by the current and previous nodes. When calculating the transition probability, instead of physical node, a memory node with historical path information will be used (we use the concepts of “physical node” and “memory node” in Rosvall’s work [34]). For example, for the path i→j→k, it can be regarded as a directed edge $\vec{ij}\to \vec{jk}$ formed by two memory nodes $\vec{ij}$ and $\vec{jk}$ in M2. The M2 dynamics can be regarded as being carried out on the network of memory nodes. On this network, the probability of state *j* transferring to state *k* is
(7)$$p\left(\vec{ij}\to \vec{jk}\right)=\frac{W(\vec{ij}\to \vec{jk})}{\sum_{l}W(\vec{ij}\to \vec{jl})}.$$

The probability of being in state $\vec{jk}$ at time t+1 is
(8)$$P\left(\vec{jk};t+1\right)=\sum_{i}P(\vec{ij};t)p\left(\vec{ij}\to \vec{jk}\right).$$

Therefore, in M2, the probability of being in state *k* at time t+1 is
(9)$$\begin{array}{c} P\left(k;t+1\right)=\sum_{j}P(\vec{jk};t+1)=\sum_{ij}P(\vec{ij};t)p\left(\vec{ij}\to \vec{jk}\right). \end{array}$$

Higher-order Markov processes are similar to second-order, in that the next state depends on the current state and several previous states. For an *m*-order Markov process, the transition probability of the next state satisfies
(10)$$\begin{array}{c} P(X_{t+1}=i_{t+1}|X_{t}=i_{t},X_{t-1}=i_{t-1},\cdots ,X_{1}=i_{1})\\ =P(X_{t+1}=i_{t+1}|X_{t}=i_{t},X_{t-1}=i_{t-1},\cdots ,X_{t-m+1}=i_{t-m+1}). \end{array}$$

For the dependency of the higher-order Markov process in the network, it is not just a simple link composed of pairwise nodes, but a path containing multiple nodes. Then, corresponding to the path i→j→⋯→k→l, recording the paths $\vec{ij\cdots k}$ as $\vec{R}$, the probability of transferring from state *k* to state *l* is
(11)$$p\left(\vec{R}\to \vec{kl}\right)=\frac{W(\vec{R}\to \vec{kl})}{\sum_{s}W(\vec{R}\to \vec{ks})}.$$

The probability of being in memory node $\vec{kl}$ at moment t+1 is
(12)$$P\left(\vec{kl};t+1\right)=\sum_{r\in \vec{R}\setminus \left\{k\right\}}P(\vec{R};t)p\left(\vec{R}\to \vec{kl}\right).$$

The probability of being in state *l* at moment t+1 is
(13)$$\begin{array}{c} P\left(l;t+1\right)=\sum_{l}P(\vec{kl};t+1)=\sum_{r\in R\setminus \{k\},l}P(\vec{R};t)p\left(\vec{R}\to \vec{kl}\right). \end{array}$$

It is worth noting that there is a distinction between higher-order Markov processes and k-step transition. The *k*-step transition probability refers to the state probability of reaching a certain state after walking *k* steps from the current state, which can be applied to the first-order Markov process or the higher-order Markov process; while the higher-order Markov process reflects the degree of dependency on the historical state.

Table 1 shows the relationship between multistep transitions and higher-order Markov processes.

## 4. Higher-Order Dependency Network

### 4.1. Concept and Introduction

Different from the FON, the nodes of the HON reflect the dependencies between the nodes under the higher-order Markov processes. In the HON, nodes with higher-order dependencies will be constructed, which is similar to the higher-order Markov processes, so we introduce them in the previous section. The node of the HON is constructed from a path consisting of current and previous nodes.

Figure 3 shows the process of transforming an FON into an HON, and the name of each node is marked next to it. Among them, the node *f* in the original FON can generate a fourth-order node at most, and f|d,c,a is a fourth-order node, indicating that the current node depends on the previous path a→c→d. f|· is a first-order node, which means that all the previous paths can become the conditions of this node. In other words, it does not depend on the previous path. f|e and f|d,c are second-order and third-order nodes.

In an HON, not all dependent paths of each node need to be taken into account unless the path appears a sufficient number of times to reduce redundancy. In the process of building an HON, the order of higher-order nodes usually does not need to reach the highest order, and the corresponding algorithm (BuildHON) [10] will determine the highest order required by a node, which will be introduced in the HON construction method in Section 4.2.

The transition probability in the previous higher-order Markov processes can be given in the form of higher-order nodes. Taking Equation (Equation 7) of the M2 as an example, it can be rewritten as follows:(14)$$\begin{array}{c} P\left(X_{t+1}=k|X_{t}=\left(j|i\right)\right)=p\left(j|i\to k\right)=\frac{W(j|i\to k)}{\sum_{l}W(j|i\to l)}. \end{array}$$

Equation (Equation 13) represents the transition probability from higher-order node j|i to first-order node *k*.

### 4.2. Construction of HON

This part refers to Xu’s [10] BuildHON algorithm consisting of two steps (**rule extraction** and **network rewiring**). We only give the general method and idea of building in our work without entering into too many details, as it is not our focus.

#### 4.2.1. Rule Extraction

Before describing the method of rule extraction, first, we explain what the rules in our work are.

There are many methods of FON storage: adjacency list, adjacency matrix, and sparse matrix derived from the adjacency matrix. A sparse matrix is represented as a series of triples.

Each triplet is in the form of <source,target,weight>, which are the source node, target node, and the weight of the edge. Weight usually refers to the transition probability between nodes. We express the triplet in another form, for example, source→target:weight. The form of this inference rule expresses the same meaning as the triple, so the FON can actually be considered as a series of rules. Reversely, the FON can be formed with these rules.

Therefore, the prerequisite for building an HON is to obtain all the rules, so the first step in building an HON is to extract rules. An HON can also be represented by a series of rules. As mentioned in Table 1, the one-step transition of a higher-order Markov process is a process from path to node; then, the source node of the higher-order rule can be expressed as a higher-order node, the target node is a first-order node, and the weight is the transition probability between nodes. It is worth noting that among the rules of an HON, there are also a large number of rules in which both the source node and the target node are first-order.

The brief steps of rule extraction are as follows:According to the given sequence, calculate the frequency of all subsequences more than the minimum support (MinSupport) from the second order to the highest order (MaxOrder).Starting from the second order, calculate the distribution frequency of the higher-order subsequence and the current subsequence.Compare whether the frequency distribution of the two sequences changes significantly, and accept the higher-order subsequence as a rule if there is a significant change; otherwise, use the current order subsequence as a rule.

The index used to compare the difference between two distributions is KL divergence, and the calculation method is
(15)$$D_{KL}(D_{2}\left|\right|D_{1})=\sum_{i\in D_{2}}P_{2}\left(i\right)\log_{2}\frac{P_{2}\left(i\right)}{P_{1}\left(i\right)}.$$

Among them, $D_{1}$ represents the probability distribution of the current order rule, and $D_{2}$ represents the probability distribution of the higher-order rule.

Define the threshold δ:(16)$$\delta =\frac{Order_{2}}{\log_{2}\left(1+support_{i\in P_{2}}\right)},$$
where $Order_{2}$ is the order of $D_{2}$ and support is the number of times the rule appears.

The higher-order sequence $S_{2}=\left[s_{t-(n+1)},s_{t-n},\cdots ,s_{t}\right]$ is grown from the lower-order sequence $S_{1}=\left[s_{t-n},\cdots ,s_{t}\right]$. Then, the number of occurrences of the higher-order sequence is $support_{i\in D_{2}}\leq support_{i\in D_{1}}$. And because of $Order_{2}=Order_{1}+1$, when the order increases, the denominator of the threshold δ decreases, the numerator increases, and the value also increases monotonically.

For $D_{KL}(D_{2}\left|\right|D_{1})$, there is
(17)$$\begin{array}{c} \max(D_{KL}(D_{2}\left|\right|D_{1}))=\max\left(\sum_{i\in D_{2}}P_{2}\left(i\right)\log_{2}\frac{P_{2}\left(i\right)}{P_{1}\left(i\right)}\right)\\ \leq 1\times \log_{2}\frac{1}{\min(P(i))}+0+0+\cdots \\ =-\log_{2}\left(\min\left(P\left(i\right)\right)\right). \end{array}$$

As the order increases, the threshold δ also increases. Therefore, when $-\log_{2}\left(\min\left(P\left(i\right)\right)\right)>\delta$, it means that the value of min(P(i)) at this time is small, which also means that the distribution is more dispersed. Therefore, when $D_{KL}(D_{2}\left|\right|D_{1})>\delta$, it can be considered that $D_{2}$ has a significant change on $D_{1}$.

When the value of KL divergence is greater than the threshold δ, the distribution of higher-order rules is considered to be significantly different from that of lower-order rules, and the higher-order rules are accepted. The rule extraction process is shown in Figure 4.

#### 4.2.2. Network Rewiring

After obtaining the rules, the steps to construct an HON are mainly:Convert all first-order rules to edges;Convert higher-order rules into edges and create corresponding higher-order nodes;Add incoming edges for higher-order nodes;The target nodes in the higher-order rules are all first-order, but if there are corresponding higher-order nodes generated by other rules at this time, the first-order target nodes can be merged into the higher-order nodes.

The network construction process is shown in Figure 5.

## 5. Experiments

### 5.1. Datasets and Design of Experiments

The datasets used in this paper are the Charades [15] and CharadesEgo [16], which come from ECCV2016 and CVPR2018, respectively. The two datasets are basically consistent in data format, so we merge them together. The merged dataset contains behavior sequence data extracted from 17,708 videos (9848 from Charades and 7860 from CharadesEgo). Charades and CharadesEgo were originally datasets for behavior detection in the field of computer vision. The two datasets provide a large number of videos and video label data for training. It is worth mentioning that the data used in our work are the labels extracted from the videos, not the videos themselves.

The dataset contains 157 behaviors, which are referred to using numbers from c000-c157. The specific behaviors corresponding to each behavior can refer to the file “Charades_v1_classes.txt”, which can be downloaded from the dataset link provided at the end of the article.

In this paper, three experiments of random walk prediction, vital node identification, and community detection are carried out on an FON and an HON, respectively. The purpose of random walk prediction is to compare the pros and cons of the HON with the FON in terms of sequence prediction accuracy. The purpose of identifying vital nodes is to explore the impact of high-order dependencies on node importance changes and to explore the rules of their changes. The purpose of carrying out the community detection experiment is to highlight the fuzziness of the community division in the HON, and to give a more optimized community affiliation bias. Among the three experiments, random walk is the basic experiment, and the remaining key node identification and community detection experiments rely on the principle of random walk, so the improvement of random walk on the HON will also affect the results of the latter two experiments’ performance. The purpose and relationship of the three experiments are shown in Figure 6:

### 5.2. Data Preprocessing

#### 5.2.1. Converting Videos to Sequence Data

In Charades and CharadesEgo, each video has an action field, which consists of behaviors extracted from the corresponding video; each behavior can be expressed as a triplet in the form of <number of behavior, start, and end >. For example: <c006, 21.80, 31.70> indicates a behavior starting at 21.80 s and ending at 31.70 s of the video. The number corresponds to the behavior description of “closing a door”.

However, in the original dataset, these behavior triplets are unordered. In order to construct a temporal behavior sequence, we sort these behavior triplets by start time. For a toy example: [<c137 0.30 9.00>, <c092 0.00 18.90>, <c152 3.50 31.00>] contains c137,c092,c152 three behaviors; the start time of these three behaviors are 0.30, 0.00, and 3.50, respectively. The toy example will be sorted into [c092,c137,c152] by start time, and now we obtain a temporal behavior sequence. The process for converting videos to sequence data is shown in Figure 7. Following this approach, we convert the 17,708 videos to 17,708 temporal behavior sequences which will be the input for the subsequent construction of the HON.

#### 5.2.2. Dividing Data by Scene

For each video, there is a main scene where the behavior occurs, which corresponds to the “scene” field in the dataset. To facilitate subsequent subscene analysis, the 17,708 temporal behavior sequences were split according to the “scene” field (15 main scenes and other scenes). The descriptive statistics are shown in Table 2.

#### 5.2.3. Network Construction

Based on the behavior sequence data, we constructed two networks, FON and HON. We added a table with basic information about the generated FON and HON. The basic attribute characteristics of the two networks are added to Table 3.

We used Gephi to visualize the obtained FON and HON, as shown in Figure 8 and Figure 9.

### 5.3. Performance of Random Walk on FON and HON

#### 5.3.1. Introduction of Experiment

Based on the temporal behavior sequences above, we constructed an FON and an HON, and conducted random walk on the two networks, respectively. The experimental results illustrate that the HON significantly improves the accuracy of random walk compared with the FON.

The detailed steps are shown in Figure 10. First, we divided the sequences satisfying certain length requirements into a training part and a test part. After that, we constructed the FON and the HON, respectively, and conducted random walk on the two graphs, respectively. Finally, we calculated the accuracy of random walk on both networks, and we found that the HON achieved an absolutely correct prediction compared to 33% from the FON. In short, the HON improved the accuracy of random walk by revealing more implied information.
(18)$$Ratio_{i}=log\frac{\alpha_{iH}}{\alpha_{iF}},i\in \left\{1,2,3\right\},$$
where *i* is the step of random walk, $\alpha_{iH}$ denotes the accuracy of *i*-step random walk in the HON, and $\alpha_{iF}$ denotes the accuracy of *i*-step random walk in the FON. $Ratio_{i}$ is the logarithm of ratio of $\alpha_{iH}$ to $\alpha_{iF}$. We can obtain the value of $\alpha_{iH}$ and $\alpha_{iF}$ by dividing the number of correct prediction by the number of test path (note that when the number of steps is greater than 1, correct prediction counts only if all steps are correctly predicted).

#### 5.3.2. Results without Division of Scenes

An FON and an HON were constructed based on the training part of the temporal behavior sequences without dividing the scenes. Then, based on the constructed networks, we performed random walk 1000 times to predict 1 step, 2 steps, and 3 steps of each temporal behavior sequence, and the prediction was correct only if the prediction was consistent with the test part. The proportions of correct predictions in 1000 random walks for 1 step, 2 steps, and 3 steps are shown in the first bar of Figure 11.

From the experimental results, we can see that the HON can significantly improve the accuracy of the random walk, and the accuracy improvement is more obvious as prediction step increases. When performing one-step prediction, the FON only considers one node before the test part, and performs a random walk based on the probability distribution of outflow from that node, which will inevitably ignore path dependencies of beyond 1 step. In the HON, these ignored path dependencies are taken into consideration, and, thus, it improves the accuracy of random walk. On the other hand, the path dependency effect of higher order is accumulated in each step, so the improvement of accuracy of the HON compared to the FON is more obvious when multistep prediction is performed.

#### 5.3.3. Results with Division of Scenes

In order to discuss the accuracy of the random walk algorithm under different scenes, we divided the entire dataset into 15 scenes (to ensure sufficient training data, we finally selected 4 scenes with more than 1000 sequences of behaviors for the experiment), and conducted random walk experiments on each scene separately. The results of the experiments are shown in Figure 11.

One-step accuracy under the FON is significantly lower than that of the HON in the Bedroom, Kitchen, and Living Room scenes. As the number of prediction steps increases to 2 or 3, the accuracy of the FON is reduced to 0. The reason for accuracy improvement on the HON compared to the FON in these three scenes is also higher-order dependencies between the temporal behavior sequences.

We note that the bars corresponding to two-step and three-step prediction in the Bedroom, Kitchen, and Living Room scenes are not shown. The reason is that accuracy based on the FON in these scenes is 0, which leads to a infinite ratio. The correct prediction of the random walk is based on the fact that the last few behavior nodes of the training part and the test part together form a pattern, and such a pattern needs to be included in the rule. This premise is easily satisfied in one-step prediction; however, when the number of prediction steps is increased to two or three steps, the premise becomes hard to satisfy.

Another noteworthy phenomenon is that in the Bathroom scene, there is no difference in the three-step accuracy between the FON and the HON, unlike the one-step and two-step experiments. We first found that in the Bathroom scene, the one-step prediction accuracy of the HON was 4.0936 times that of the FON, and the two-step prediction accuracy was 13.4664 times, both of which were lower compared to the other scenes, which could prove that higher-order dependencies between behaviors in the Bathroom scene were weaker than other scenes. In addition, unlike the other three scenes, the highest order of the HON constructed in the Bathroom scene is only 2-order, i.e., there is no 3-order dependency in this subdataset, which means the HON does not have a significant advantage over the FON when performing the three-step prediction, and finally means that the three-step prediction accuracy of the FON and the HON is very close.

#### 5.3.4. Analysis of Results

**The above results show that the accuracy of random walk prediction is significantly improved regardless of whether the scenes are divided or not.** In the process of HON reconstruction, the large number of higher-order rules generated shows that there are more higher-order dependencies among the behavior sequences. This indicates that there are dependencies in the occurrence of user behaviors, and the requirements linked to the relevant behaviors are equally dependent and traceable. The accuracy of random walk in the HON is significantly higher than that of the FON, which indicates that higher-order dependencies play a greater role in the prediction of behaviors. Therefore, the accuracy improvement in higher-order networks may be more significant when using path prediction algorithms derived from random walk methods. In the behavioral analysis, the accuracy improvement of behavioral sequence prediction brought about by the HON will be more accurate in predicting and mining requirements, thus providing more accurate personalized recommendations. In the intelligent home scenario, relying on IoT, accurate behavioral prediction and requirements mining can also provide better quality of life services.

### 5.4. Performance of PageRank on FON and HON

#### 5.4.1. Introduction of Experiment

Based on the constructed behavior network, we can use the PageRank algorithm to identify high-frequency behaviors of indoor life. The PageRank algorithm is based on random walk, and the HON also significantly changes the results of the PageRank algorithm [38] by improving the accuracy of random walk.

We adopted a weighted PageRank algorithm with random jump. And for node *v*, its score at time t+1 is given by
(19)$$PR_{t+1}\left(v\right)=\frac{1-\beta }{N}+\beta \cdot \sum_{u\in B(v)}\left(\frac{w_{uv}}{w_{u}}\cdot PR_{t}\left(u\right)\right).$$
where the score of node *v* at time t+1 consists of two parts, one from random jump and the other from neighbors B(v) to node *v*, and the proportion of the two parts is adjusted by parameter β (in our work, β=0.85). When β is small, there will be greater probability for the walker to jump from the current node. Otherwise, the probability for the walker to jump will be smaller. In Formula (Equation 19), $PR_{t+1}\left(v\right)$ denotes the score of node *v* at time t+1, and $PR_{t}\left(u\right)$ denotes the score of node *u* at time *t*, where there is an edge from node *u* to node *v*, i.e., u∈B(v). *N* denotes the number of nodes of the network. $w_{u}$ is the sum of weights of edges starting from node *u*, in which $w_{uv}$ is the sum of weights of edges from node *u* to node *v*.

First, we construct the FON and the HON separately. Then, the PageRank algorithm is applied to calculate the score of each node separately. In the HON, the score of the higher-order node is the score for the entity node, i.e., the sum of all state nodes, which is shown in Figure 12.

#### 5.4.2. Results of Experiment

The nodes are ranked on the FON and the HON without dividing the scenes. The probability of random jumping in the PageRank algorithm is set to 0.15, the convergence criteria is set to $10^{-9}$, and the maximum number of iterations is 1000. And Table 4 shows the top three behaviors with large score change (score change = HON score of node – FON score of node).

Table 4 lists the three behavior nodes with the largest positive change and negative change in score. The three behavior nodes with the largest increase in score during the change from FON to HON are c156, c061, and c065, and the nodes with the largest decrease in score are c151, c152, and c033.

Each node name represents a behavior; the three nodes with the highest score increases were *someone is eating something*(c156) (0.015798 to 0.034973), *holding some food*(c061) (0.022992 to 0.040868), and *eating a sandwich*(c065) (0.007647 to 0.020194). The three nodes with the most decreasing scores are *someone is going from standing to sitting*(c137) (0.013632 to 0.00861), *someone is smiling*(c125) (0.022617 to 0.015149), and *holding a towel*(c038) (0.02568 to 0.017065).

#### 5.4.3. Analysis of Results

In Table 4, we rank the top three discrepancies, in terms of weight, for the node with the most increased score (c156) and the node with the most decreased score (c151).

In Figure 13, we show the change of inflow and outflow from the FON to the HON for node c156 and node c151, respectively. The height of the bars indicates the sum of weight, which is given by
(20)$$\begin{array}{c} H_{in}=\sum_{a\in B(b)}w_{a\to b}^{H}-\sum_{a\in B(b)}w_{a\to b}^{F}\\ H_{out}=-\left(\sum_{c\in A(b)}w_{b\to c}^{H}-\sum_{c\in A(b)}w_{b\to c}^{F}\right) \end{array}$$
where node *b* can be node c156 or c151. B(b) is the set of nodes before node *b*, i.e., there is an edge from node *a* to node *b*. And A(b) is the set of nodes after node *b*, i.e., there is an edge from node *b* to node *c*. We sum the weight of edges flowing in node *b* in HON, and minus the sum in FON, then obtain the height $H_{in}$ of the bar named “in”. The height of $H_{out}$ can be obtained in the same way except that $H_{out}$ should be a negative value.

From FON to HON, the scores of nodes change a lot and we take a close look at node c156 and node c151 which are the nodes that gain the most and lose the most, respectively. The inflow of node c156 increases by 158 units, and its outflow increases by 84 units, which results in a 74-unit net inflow, and we fold the bar of outflow increment onto the bar of inflow increment to show the net inflow prominently. For node c151, by folding a inflow increment of 17 units into a outflow increment of 140 units, we obtain a 123-unit net outflow.

In Figure 14, we present a marginal histogram to analyze the difference of score for the FON and the HON. For every node, its coordinate on the *y*-axis indicates the score in the HON, and coordinate on the *x*-axis indicates the score in the FON. The nodes on the curve in royal blue satisfy “y=x”, which means that their coordinates on the *x*-axis are equal to those on the *y*-axis. The red curve is the fitted curve of scatters. We find that the fitted curve deviates far from the curve “y=x”. The slope of the fitted curve is larger than 1 and it has a negative bias. This means that the nodes with a low score in the FON will obtain an even lower score in the HON, and nodes with a high score will obtain a higher score in the HON. In other words, **when we rank nodes with the PageRank algorithm in the process of changing from the FON to the HON, the strong become stronger and the weak become weaker**. A word of caution: this phenomenon does not occur on every node as it is statistical. In the upper and right part of Figure 14, we show the normal distributions of scores in the FON and the HON. It is obvious that the distribution curve of the HON is flatter than that of the FON, which reveals a more significant distinction between nodes with low score and nodes with high score in the process of changing from the FON to the HON, in accordance with the phenomenon indicated by the fitted curve.

#### 5.4.4. Comparison of Different Vital Node Identification Algorithms

We continued to use LeaderRank [39], Hits [40], and Eigenvector Centrality [41], three vital node identification algorithms applied to the FON and the HON. The maximum number of iterations of the above algorithm is 1000, and the tolerance is 0.00005. The five nodes with the highest score among the FON and the HON for each algorithm are listed in Table 5.

The three new algorithms were applied to the HON and the FON, respectively, and the five behaviors with the highest scores are listed in the table. From the comparison of performance on the two networks, the difference between **Hits** on the FON and the HON is the largest, and the behavior of the top five has changed greatly, while the performance difference of **Eigenvector Centrality** on different networks is the smallest. From the comparison between algorithms, the performance of **PageRank**, **Eigenvector Centrality**, and **LeaderRank** on the FON is similar, while the performance of **Hits** is quite different from the first three algorithms. On the HON, the differences between the four algorithms generally become larger.

For the behavior sequence network in this paper, it is not possible to judge the performance of different algorithms, but different algorithms show different evaluation characteristics. However, after comparative experiments, we found that for node importance evaluation on the FON, the performance difference between different algorithms is relatively small; while on the HON, the difference is relatively large. This shows that the complexity of the HON amplifies the difference between nodes of importance. The identification of vital nodes through multiple algorithms can also increase the credibility of some nodes with high scores (such as c154 and c151).

### 5.5. Performance of Infomap on FON and HON

#### 5.5.1. Introduction of Experiment

Infomap is an algorithm for community detection based on random walk, and it has been used in overlap community detection [42,43]. Infomap combines community detection and information coding. The main idea of Infomap is that a good division of communities has a short code of sequence sampled from a network.

Firstly, we sampled a sequence consisting of nodes (just like in PageRank; there is also a probability to jump from the current node when sampling nodes). After that, we clustered nodes based on double-layer coding to make the sequence as short as possible. The main idea of double-layer coding is to number every community and also number every node in the same community. To achieve this, we can identify every node in network uniquely in the form of “community number + node number”.

For a scheme of encoding,
(21)$$codePlan=\{c_{1},c_{2},...,c_{m}\},$$
where all nodes are divided into *m* communities $c_{1},c_{2},...,c_{m}$. A community $c_{i}$ including $n_{i}$ nodes is given by
(22)$$c_{i}=\left\{v_{1},v_{2},...,v_{n_{i}}\right\},i\in \left\{1,2,...,m\right\}.$$

When sampling a node by random walk on a network with *n* nodes, with probability ε of jump from the current node, the probability of visiting node $v_{\beta }$ is given by
(23)$$p_{v_{\beta }}=\sum_{v_{\alpha }\in B\left(v_{\beta }\right)}p_{v_{\alpha }}\cdot \left[\left(1-\varepsilon \right)p_{v_{\alpha }\to v_{\beta }}+\frac{\varepsilon }{n}\right].$$
where node $v_{\alpha }$ has an edge to $v_{\beta }$, i.e., $v_{\alpha }$ is one of the nodes in neighbors $B(v_{\beta })$.

For each sequence sampled from a network, its length of code is equal to length of community number plus length of node number. The expected value of length of sequence code is given by
(24)$$L_{total}=p_{jump}\cdot L_{jump}+p_{inter}\cdot L_{inter}.$$

$L_{total}$ is the average length of code for sequence sampled, in which $p_{jump}\cdot L_{jump}$ is the average length of code for communities and $p_{inter}\cdot L_{inter}$ is the average length of nodes inside communities. The probability of jumping from the k−1-th node to the *k*-th node in sequence across two communities is given by
(25)$$p_{jump}=\sum_{s_{k-1}\in c_{i}}\sum_{s_{k}\notin c_{i}}p_{s_{k-1}}\cdot \left[\left(1-\varepsilon \right)p_{s_{k-1}\to s_{k}}+\frac{\varepsilon }{n}\right].$$

And the probability of transferring from the k−1-th node to the *k*-th node in sequence inside a community is given by
(26)$$p_{inter}=\sum_{s_{k-1}\in c_{i}}\sum_{s_{k}\in c_{i}}p_{s_{k-1}}\cdot \left[\left(1-\varepsilon \right)p_{s_{k-1}\to s_{k}}+\frac{\varepsilon }{n}\right].$$

Then, we tried sending every node to communities nearby and we saved the change in which the length of code for sequence sampled declined fastest. We repeated the program until the length of code became stable.

Compared with network detection on FONs, HONs are able to split a single node into multiple higher-order nodes, which can be divided into multiple communities, and finally the higher-order nodes representing the same original node are merged to obtain the community detection results under the HON. The calculation method of the original node’s PageRank score is shown in Figure 15.

The result of community detection is to define the affiliation relationship of nodes to communities. In a behavior network, a community may refer to a collection of behaviors that occur in succession. In the community detection in an FON, a behavior belongs to only one community, which means it is included in a single behavior set. But many behaviors appear in more than one behavior set. For example, two behavior sets { *making a sandwich*, *holding a sandwich*, *eating a sandwich* } and { *taking a sandwich from somewhere*, *holding a sandwich*, *putting a sandwich somewhere* } both have the behavior of holding a sandwich. In the community division based on an FON, this behavior can only belong to one community, but this is unreasonable. On an HON, this behavior can belong to multiple communities at the same time, that is, a collection of behaviors. Therefore, the significance of using an HON for community division is to divide behavior sets more accurately, so as to better analyze behavior rules.

#### 5.5.2. Results without Division of Scenes

The FON contains 157 nodes and the HON contains 438 nodes (157 nodes after merging), and community detection is conducted on the FON and the HON, respectively. The final results are obtained in Table 6.

The number of communities is 17 in FON, and the number of communities in HON is 26. The average community sizes are 9.24 and 16.85 nodes, respectively. A interesting phenomenon emerges on the HON regarding the number of communities which the nodes belong to. About one-fifth of the nodes in HON are divided into multiple communities, in which half of the multi-community nodes are the member of two communities, and the other half of the multi-community nodes are contained in three or more communities, with the node c059 even belonging to eight communities. In contrast, the node c119 belongs to only one community. We filter out the higher-order dependency rules in which node c059 and node c119 are the source nodes, and find that there are 24 second-order rules starting from c059 and only 2 second-order rules starting from c119. The phenomenon indicates that nodes with more higher-order dependency rules are more likely to be divided into multiple communities. It is worth noting that for a node with higher-order dependencies, it is not necessarily divided into multiple communities, because the higher-order nodes may be divided into the same community, which does not increase the number of communities the original node belongs to.

#### 5.5.3. Results with Division of Scenes

We construct an FON and an HON for four scenes, Bathroom, Bedroom, Kitchen, and Living Room, and conduct community detection separately. Table 7 reports the results of community detection in the four scenes.

Table 7 shows the results of the community detection after dividing scenes. To analyze the relationship between higher-order dependency rules and community detection, the number and percentage of higher-order dependency rules in the four scenes are also listed in the table. It is worth noting that when community detection is conducted on the HON, higher-order nodes are used.

In Bathroom, there is no significant difference in all indicators between the FON and the HON, which means that the network in the Bathroom scene does not have a significant higher-order dependency effect. Higher-order dependency rules account for 13.04% of all rules. In contrast, in the scenes of Bedroom, Kitchen, and Living Room, there are more communities in the HON than in the FON, and the community assignment is also greatly increased. In these three scenes, the proportion of higher-order rules is about 25% (much higher than the Bathroom scene).

We further verify that the proportion of higher-order dependency rules will affect the community detection results of the HON. On the one hand, the increase in the proportion of higher-order dependency rules will significantly increase the number of communities and community assignments. On the other hand, the increase in the proportion of higher-order dependency rules may also make the community smaller. Generally speaking, the increase of the proportion of higher-order dependency rules will make the community division more detailed, which is manifested in the increase of the number of communities and the decrease of community size.

#### 5.5.4. Analysis of Results

**The community detection results for HON indicate that the higher-order nodes of one original node may belong to different communities**. In behavior analysis, some behaviors may form a behavior groups, and behaviors in the same group have stronger dependencies and often occur in a certain event together. In an FON, each node belongs to only one community, while a behavior may occur in many events in real life. For example, the action c059 (sitting in a chair), which belongs to a number of communities, can occur in multiple life events, such as a meal in restaurants, relaxing in the living room, or reading in a study. The community detection for an HON allows nodes to be contained in more communities and thus more accurately assigns behaviors to a group.

#### 5.5.5. Comparison of Different Community Detection Algorithms

We continued to add Louvian [44] and Greedy Modularity [45,46], two experiments on community detection algorithms applied to directed graphs, and compared their performance in the FON and the HON. In addition to the original Infomap algorithm, the situation of the three algorithms to divide the community is shown in Table 8.

Judging from the comparison of performance on the two networks, the three algorithms all divide more communities in the HON, and the size of a single community is also larger. And due to the characteristics of the HON, a node can also belong to more communities, so the community assignment of the three algorithms in the HON is also higher. Judging from the characteristics of the three algorithms, the performance of **Louvian** is different from the other two algorithms. The community size on the HON is smaller than that on the FON, and the community assignment of the HON is also significantly higher than the other two algorithms. Because the **Louvian** algorithm divides more communities, the community size is generally small, but the community assignment increases significantly. The Greedy Modularity algorithm has a small number of divided communities, so the size of each community is also greatly increased.

For the behavior sequence network in this paper, we cannot judge the performance of different community detection algorithms. However, from the experiments of the three algorithms on the two networks, there are still some common features. Due to the increase in the number of nodes in the HON, various algorithms can divide more communities, but community size generally decreases. In the HON, the product of community assignment and number of nodes, which is fixed, is proportional to the product of community size and number of communities. Through the comparison of various algorithms, the characteristic differences between the HON and the FON in community detection are further highlighted.

### 5.6. Summary and Analysis of Experimental Results

Based on the above three experiments, we study the difference between the HON and the FON on some classic network tasks. The results of three experiments demonstrate some properties of the HON compared to the FON:Higher-order dependencies cause the network to have more accurate state transitions, and all network algorithms based on state transition will benefit.Higher-order dependencies can change the importance of nodes in the network, which can eliminate some errors caused by inappropriate first-order connections.Higher-order dependency changes the membership of nodes to communities, which is different from the single membership of nodes and communities in the FON. After considering higher-order dependency, nodes have multiple membership.

Applying higher-order dependencies to daily behavior networks brings improvements in the analysis of daily behavior. Firstly, the prediction of future behavior through random walks on the network is more accurate. Secondly, the change in the value of node importance reduces the importance of some intermediary behaviors and increases the importance of scene-specific behaviors, so it helps to identify key behaviors exclusive to some scenarios and reduces the effect of some recurring behaviors. Thirdly, multiple communities to which the behavior belongs makes it possible to effectively detect behaviors that occur in multiple scenarios when the community is classified as a scene.

## 6. Summary and Discussion

We carried out a series of experiments on behavior sequences on an HON, which basically confirmed our previous conjecture. That is to say, there are higher-order dependencies between daily life behaviors that can help to perform better on various tasks, such as identification of vital nodes and community detection of behaviors, which are based on the improvement of random walk accuracy.

The improvement of random walk accuracy is significant. There are a mass of algorithms based on random walk whose accuracy will grow significantly, for example, DeepWalk [47], node2vec [48], and so on. We can predict behavior more accurately and provide support for more accurate requirement mining and personalized recommendation. The improvement of accuracy of vital node identification can help us find out the key behaviors with high frequency, which can be used to generate more practical user behavior guidelines. The diversity of community detection shows that a behavior (especially at the intersection) belonging to only one community in an FON can be divided into multiple communities in an HON, which will reveal that the behavior is related to multiple scenes.

It is worth mentioning that the random walk algorithm used in our work for predicting behavior is not an efficient prediction algorithm. Although the accuracy in the HON is substantially better than that in the FON, the absolute value of the accuracy in the HON is still quite low (less than 10% in one-step prediction). Therefore, more efficient prediction methods can be considered in subsequent studies. In addition, the study in this paper uses sequence data without timescale, and only the order of the behaviors is considered without considering the time, which will result in the loss of more detailed information. In subsequent studies, we can try a temporal network on the tasks. The method used in this article can be promoted in many fields, such as software development process and supply chain management.

## Figures and Tables

**Figure 1 entropy-25-01120-f001:**
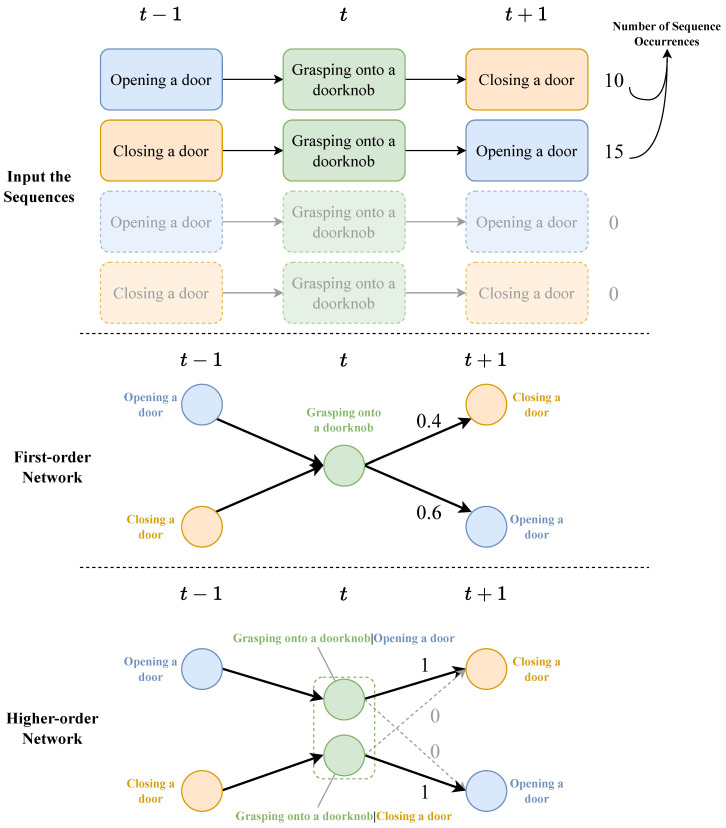
The example considers two sequences consisting of three daily actions: **opening a door**, **grasping onto a doorknob**, and **closing a door**. The two sequences describe the actions of t−1, *t*, and t+1 moments. The number after the sequence is the number of occurrences of the sequence.

**Figure 2 entropy-25-01120-f002:**
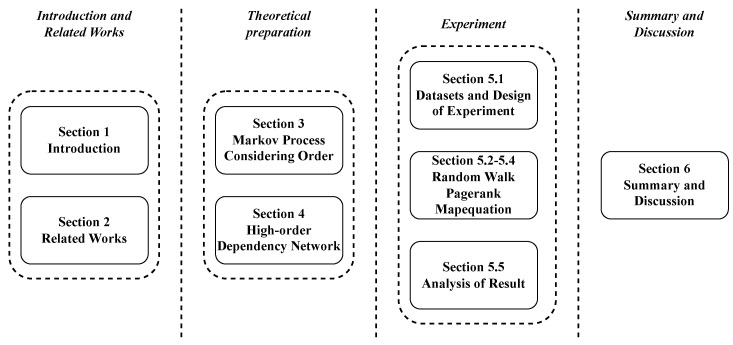
The structure of the paper.

**Figure 3 entropy-25-01120-f003:**
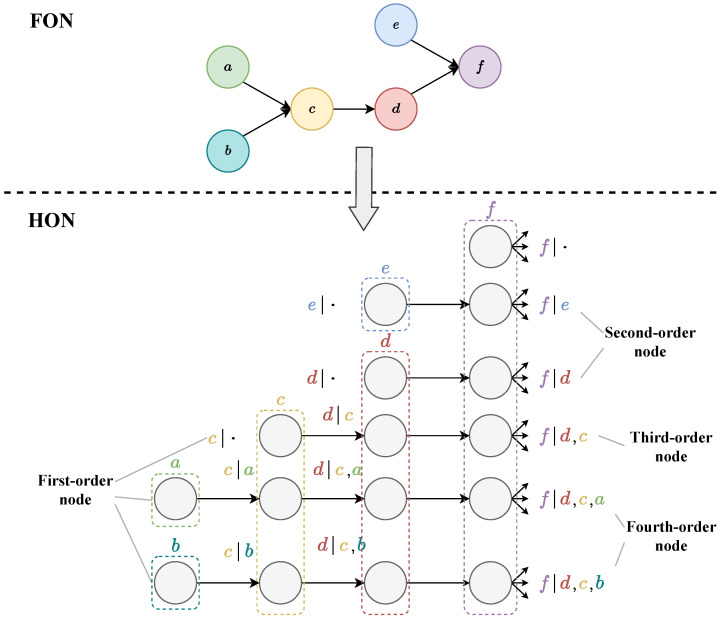
Transform an FON to an HON.

**Figure 4 entropy-25-01120-f004:**
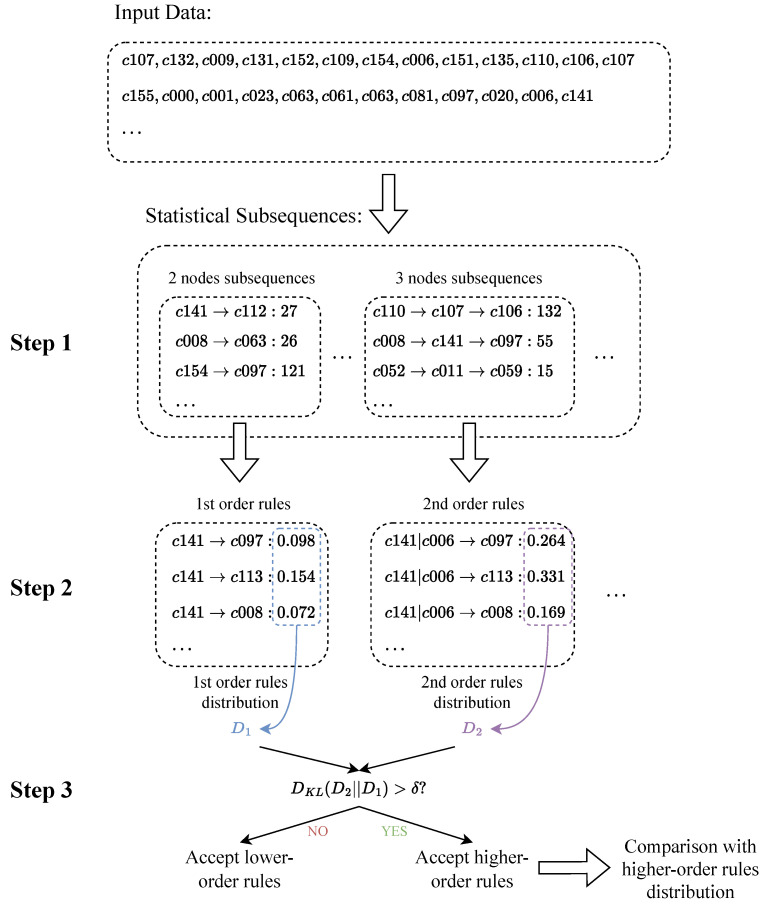
Extract rules from sequence data.

**Figure 5 entropy-25-01120-f005:**
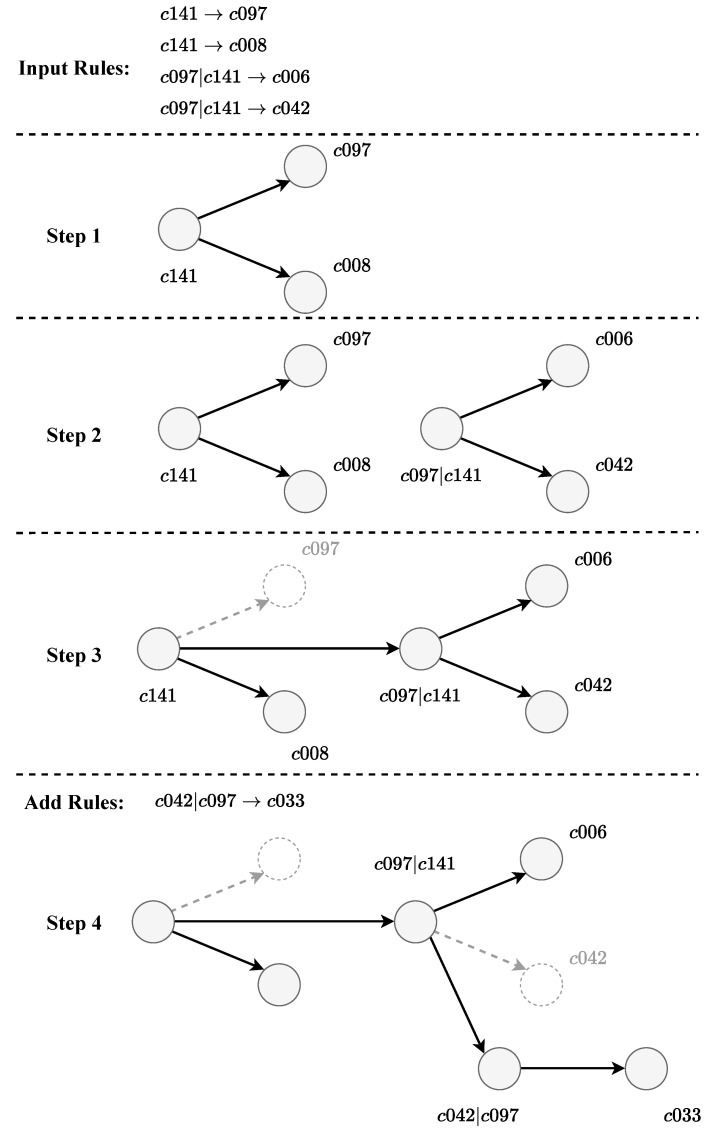
Rewire network using rules.

**Figure 6 entropy-25-01120-f006:**
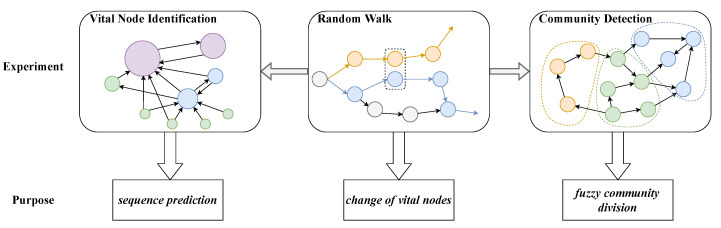
Design and relationship of three experiments.

**Figure 7 entropy-25-01120-f007:**
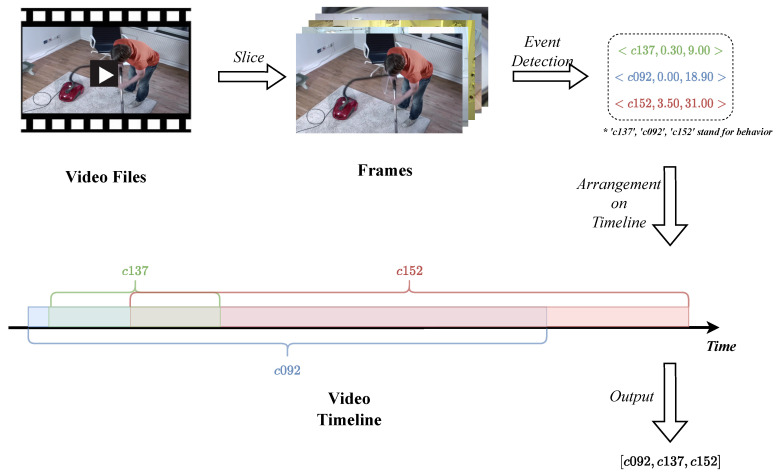
Converting videos to behavior sequences.

**Figure 8 entropy-25-01120-f008:**
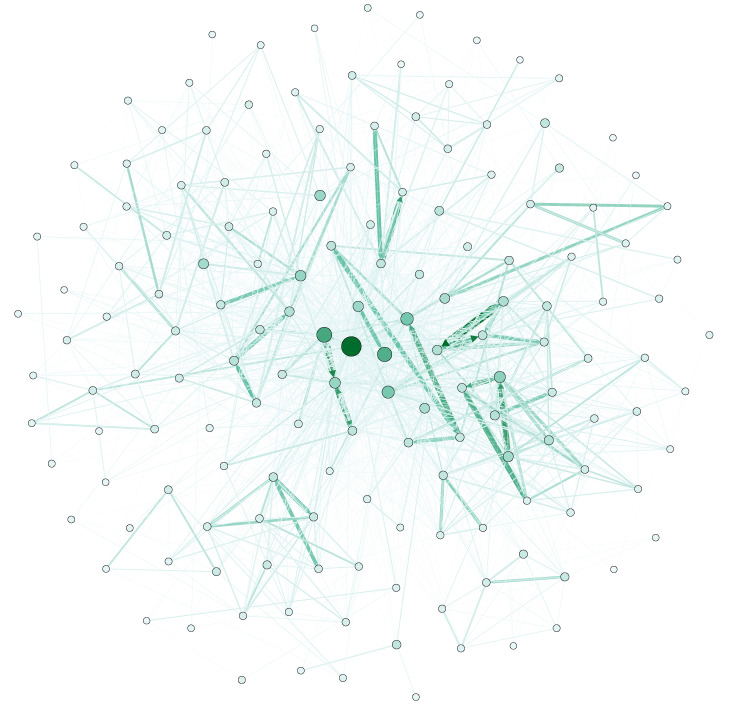
Demonstration of the FON constructed from behavioral data.

**Figure 9 entropy-25-01120-f009:**
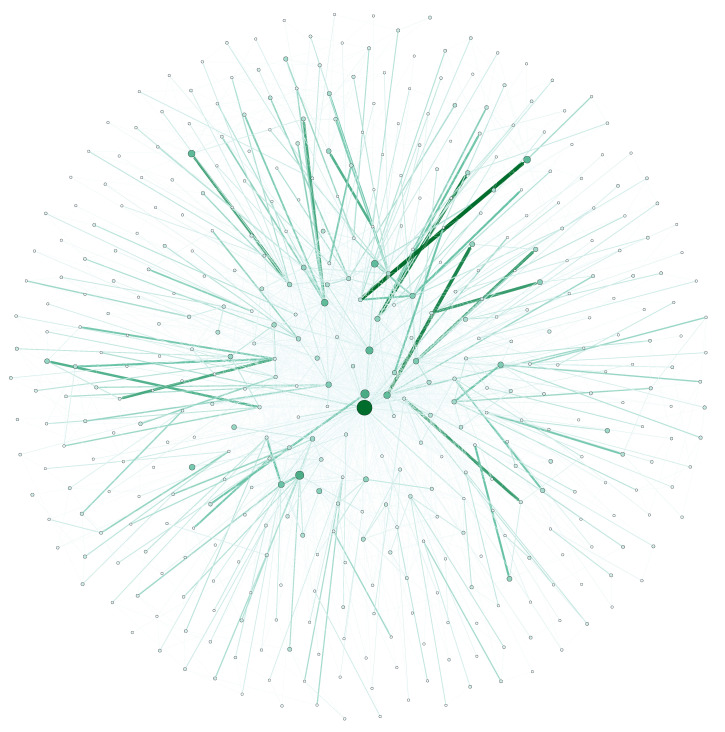
Demonstration of the HON constructed from behavioral data.

**Figure 10 entropy-25-01120-f010:**
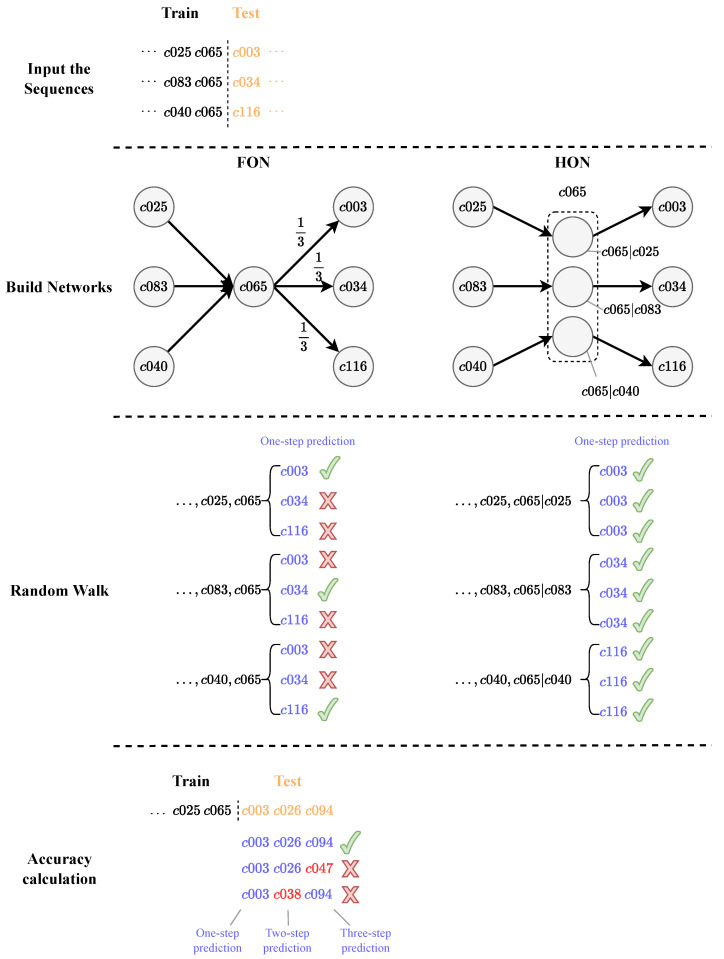
Random walk and accuracy on the FON and the HON.

**Figure 11 entropy-25-01120-f011:**
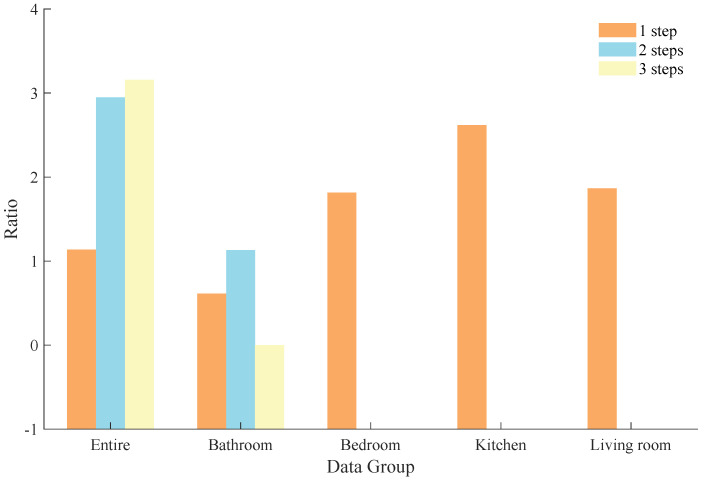
Comparison of prediction accuracy between FON and HON. The bars that are not shown represent infinite value (2-step and 3-step accuracy in Bedroom, Kitchen, and Living Room).

**Figure 12 entropy-25-01120-f012:**
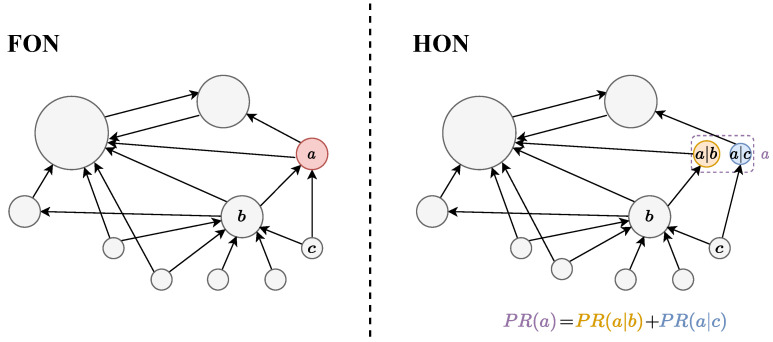
PageRank score calculation on the FON and HON.

**Figure 13 entropy-25-01120-f013:**
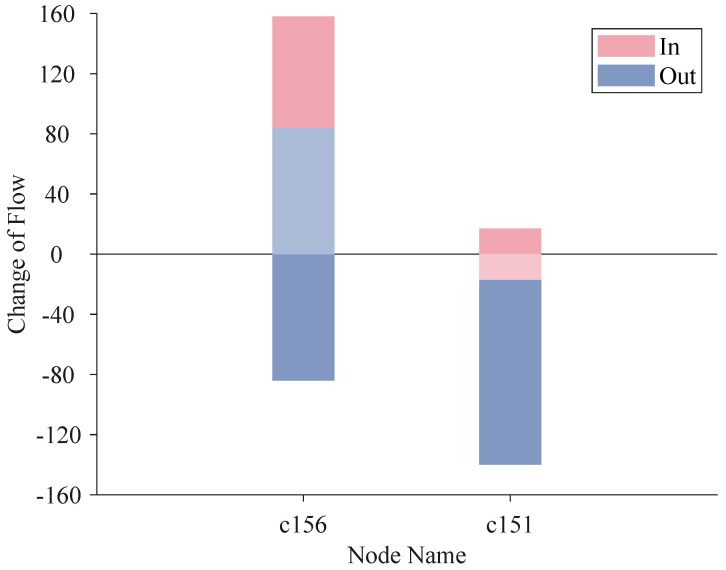
Flow change of node c156 and c151.

**Figure 14 entropy-25-01120-f014:**
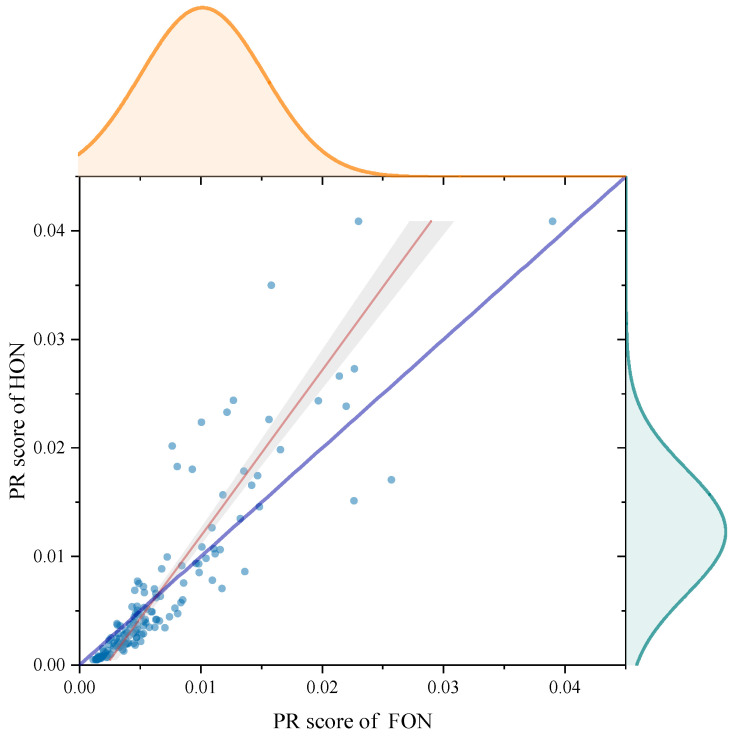
PageRank score analysis in the FON and the HON.

**Figure 15 entropy-25-01120-f015:**
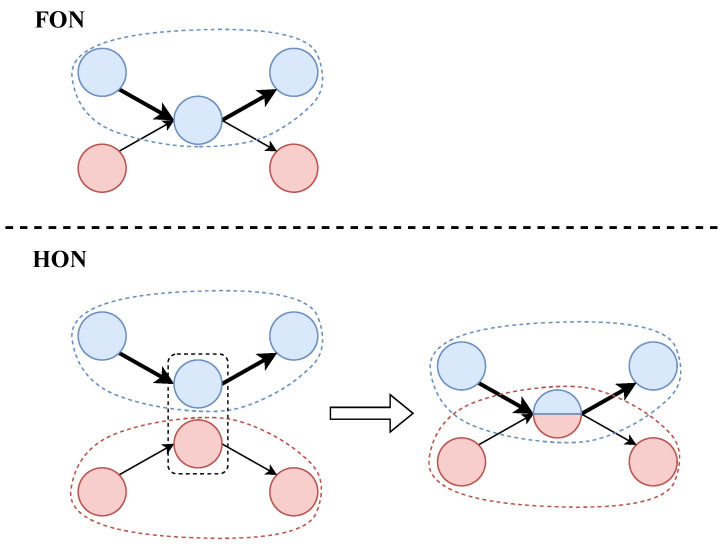
Community detection in the FON and the HON.

**Table 1 entropy-25-01120-t001:** The relationship between higher-order Markov and multistep transition probability.

		Markov Process Order	
		First-Order	Higher-Order
**Transfer steps**	**One-step**	Node to node	Path to node
	**Multistep**	Node to path	Path to path

**Table 2 entropy-25-01120-t002:** Descriptive statistics of subdatasets divided by scenes.

Scene	Size of Dataset	Average Length of Sequence
Basement	250	7.27
Home office	964	8.65
Bathroom	1144	7.15
Kitchen	3078	8.86
Bedroom	3213	9.08
Laundry room	727	8.23
Closet	680	8.61
Living room	2815	9.12
Dining room	925	9.12
Pantry	449	8.54
Entryway	882	8.48
Recreation room	462	8.60
Garage	378	7.14
Stairs	755	7.65
Hallway	775	8.14
Other	211	8.26

**Table 3 entropy-25-01120-t003:** Network descriptive statistics.

Attributes	FON	HON
Number of nodes	157	438
Number of edges	2642	3107
Average degree	16.828	7.094
Average weighted degree	497.548	196.153
Density	0.108	0.016

**Table 4 entropy-25-01120-t004:** Top 3 positive and negative score change of nodes.

Node Name	Change of Score	Node Name	Change of Score
c156	$1.91\times 10^{-2}$	c151	$-8.62\times 10^{-3}$
c061	$1.79\times 10^{-2}$	c152	$-7.47\times 10^{-3}$
c065	$1.25\times 10^{-2}$	c033	$-5.02\times 10^{-3}$

**Table 5 entropy-25-01120-t005:** Ranking of top 5 nodes under different vital node identification algorithms.

Rank	PageRank		LeaderRank		Hits		Eigenvector Centrality	
	FON	HON	FON	HON	FON	HON	FON	HON
1	c154	c154	c154	c154	c107	c059	c154	c154
2	c151	c061	c097	c059	c061	c154	c151	c151
3	c061	c156	c151	c151	c059	c107	c152	c152
4	c152	c009	c059	c097	c106	c106	c009	c009
5	c009	c059	c152	c152	c154	c151	c097	c061

**Table 6 entropy-25-01120-t006:** Result comparison of community detection.

	Number of Community	Community Size	Community Assignment	Size of Largest Community
FON	17	9.24	1	27
HON	26	16.85	1.39	40

**Table 7 entropy-25-01120-t007:** Results of community detection of divided scenes.

	Bathroom		Bedroom		Kitchen		Living Room	
	FON	HON	FON	HON	FON	HON	FON	HON
Number of community	19	23	15	58	9	64	2	70
Community size	7.11	9.43	10.4	13.28	17.44	18.05	78.5	20.07
Community assignment	1	1.26	1	3.71	1	5.05	1	5.73
Size of largest community	19	24	42	53	68	77	148	103
Number of higher -order rules	-	88	-	764	-	650	-	624
Percentage of higher -order rules	-	13.04%	-	25.38%	-	23.78%	-	24.22%

**Table 8 entropy-25-01120-t008:** Results of different community detection algorithms.

	Infomap		Louvian		Greedy Modularity	
	FON	HON	FON	HON	FON	HON
Number of community	17	26	22	63	6	9
Community size	9.24	16.85	7.48	7.18	26.17	48.67
Community assignment	1	1.39	1	2.51	1	1.25
Size of largest community	27	40	18	25	53	113

## Data Availability

The common dataset used in this study can be obtained from https://prior.allenai.org/projects/charades and https://prior.allenai.org/projects/charades-ego (all accessed on 14 June 2023).

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
