# Peer review of "Research on User Behavior Based on Higher-Order Dependency Network"

_entropy, 2023, doi:10.3390/e25081120_

Round 1

Reviewer 1 Report

In this paper, high-order dependency networks are used to conduct research on user behavior data. The article carried out three experiments of random walk, vital node identification and community detection on the high-order dependency network. Compared with the first-order network, the results of the three experiments were successfully improved. Regarding this article, the following suggestions and questions are raised:

1. In the related work, please further explain the relationship between higher-order networks and higher-order dependency networks. Please make some distinction between the two and introduce them separately.

2. What is the connection and relationship between Markov processes considering order and higher-order dependency networks, and why do we need to include a section on Markov processes considering order?

3. A total of three experiments are given in the experimental section. Please explain why these three experiments were conducted and how they are related and connected to each other.

4. The article provides a specific analysis of the results for each experiment. However, it lacks an overall analysis of the results and the significance of the experiment after the three experiments were completed. Please add this part.

5. The significance of community detection for action sequence analysis is unknown. Please give a specific example of what community detection actually means for action sequence analysis.

6. Basic descriptive statistics for higher-order dependency networks and traditional networks should be included.

This article still has some grammatical errors and needs further revision.

Author Response

On behalf of my co-authors, we thank you very much for giving us an opportunity to revise and submit our manuscript. We appreciate the editor and reviewers very much for their positive and constructive comments and suggestions on our manuscript entitled “Research on User Behavior Based on Higher-Order Dependency Network” (entropy-2488057). We have placed the specific modification instructions in the attached document.

Reviewer 2 Report

1.        The authors adopt indoor daily behavior sequences obtained by video behavior detection, extracts higher-order dependency rules from behavior sequences, and used RandomWalk algorithm. Therefore, what is the technology contribution in this study?

2.        How to define reasonable depth in the high-order network?

3.        From the rule extraction given in this study, how to give the reasonable threshold delta?

4.        In the section of experiment, please compared with recent competition methods.

   Readability should be improved. The manuscript should be further improved by English native speakers, especially for those grammatical errors, typographical errors.

Author Response

(The authors gave the same response as above.)

Round 2

Reviewer 1 Report

The authors revised this paper accoeding to my suggestions, I suggest to ACCEPT this paper.

It is okay.